# Anti-Inflammatory Action and Mechanisms of Resveratrol

**DOI:** 10.3390/molecules26010229

**Published:** 2021-01-05

**Authors:** Tiantian Meng, Dingfu Xiao, Arowolo Muhammed, Juying Deng, Liang Chen, Jianhua He

**Affiliations:** 1College of Animal Science and Technology, Hunan Agricultural University, Changsha 410128, China; meng-tiantian@foxmail.com (T.M.); mbayor88@gmail.com (A.M.); djuyingyx@foxmail.com (J.D.); 2Huaihua Institute of Agricultural Sciences, No.140 Yingfeng East Road, Hecheng District, Huaihua 418000, China; chenliang890709@163.com

**Keywords:** resveratrol, absorption and metabolism, anti-inflammation, antioxidant, mechanism

## Abstract

Resveratrol (3,4′,5-trihy- droxystilbene), a natural phytoalexin polyphenol, exhibits anti-oxidant, anti-inflammatory, and anti-carcinogenic properties. This phytoalexin is well-absorbed and rapidly and extensively metabolized in the body. Inflammation is an adaptive response, which could be triggered by various danger signals, such as invasion by microorganisms or tissue injury. In this review, the anti-inflammatory activity and the mechanism of resveratrol modulates the inflammatory response are examined. Multiple experimental studies that illustrate regulatory mechanisms and the immunomodulatory function of resveratrol both in vivo and in vitro. The data acquired from those studies are discussed.

## 1. Introduction

In 1976, resveratrol (3,4′,5-trihy- droxystilbene) was thought to be just a phytoalexin [1], one of the polyphenolic compounds produced by plants in response to environmental stress [2]. Subsequently, resveratrol was found to exhibit multiple bioactivities, including anti-oxidative [3], anti-inflammatory [4], cardiovascular protective [5], and anti-aging [6] properties, in animals. This phytoalexin has been found in at least 72 plant species, such as mulberries, peanuts, and grapes (Table 1). As a nonflavonoid polyphenol, resveratrol exists as two geometric isomers, trans- and cis- (Figure 1), and their glucosides, trans- and cis- piceids [7].

Since there may be important clinical implications for the positive roles of resveratrol on inflammatory response control, the aim of this article is to examine its strong anti-inflammatory action and the potential molecular mechanisms of such effects.

## 2. Absorption and Metabolism of Resveratrol

Previous studies have characterized in vitro and in vivo resveratrol absorption rates [8,9]. Resveratrol is absorbed in large quantities by enterocytes after oral administration [10], but only a small part of this compound ingested from the diet reaches the bloodstream and body tissues [9]. Additionally, owing to the complex structure and high molecular weight, the well-operating metabolism of resveratrol in the liver and intestine leads to oral bioavailability of about 12% of trans-resveratrol [8].

Resveratrol is absorbed by passive diffusion or carrier-mediated transport across the enterocyte apical membrane, and then rapidly and extensively metabolized to resveratrol glucuronides or sulfates [11]. Meanwhile, a significant portion (about 90%) of ingested resveratrol reaches the colon in their intact form and is subsequently subjected to gut fermentation. Once absorbed through the portal vein, the produced polyphenolic metabolites enter the liver where they are further methylated, glucuronidated or sulfated. Then, the metabolites will penetrate into the systematic circulation and reach the target tissues and cells where physiological significance can be demonstrated. Resveratrol and unused metabolites can be recycled back to the small intestine through the bile or excreted through urine [9].

Rapid absorption, poor bioavailability, and low aqueous solubility are some of the crucial limitations and challenges of the in vivo use of resveratrol [12,13]. Therefore, different methodological approaches, such as solid lipid nanoparticles (SLNs) and nanostructured lipid carriers (NLCs), have been used to improve the poor aqueous solubility and the low bioavailability of resveratrol [14].

## 3. The Anti-Inflammatory Activity of Resveratrol

Inflammatory response is a multi-stage process involving multiple cell types as well as mediator signals [15]. Inflammation is an adaptive response, which can be triggered by various danger signals, such as invasion by microorganisms or tissue injury [16]. The exogenous and endogenous signaling molecules are known as pathogen-associated molecular patterns (PAMPs) and damage-associated molecular patterns (DAMPs), respectively [15,16,17].

Both PAMPs and DAMPs are recognized by various pattern recognition receptors (PRRs), such as Toll-like receptors (TLRs) [18,19,20]. The activation of PRR induces intracellular signaling cascades, such as kinases and transcription factors [19,21]. The signaling pathways mentioned above can promote the production of a variety of inflammatory mediators (such as cytokines) for inflammation development.

Multiple lines of evidence from laboratory studies, both in vivo and in vitro, have shown that the anti-inflammatory properties of resveratrol may be explained though inhibiting the production of anti-inflammatory factors. For example, resveratrol has been stated to suppress the proliferation of spleen cells induced by concanavalin A (ConA), interleukin (IL)-2, or allo-antigens, and to more effectively prevent lymphocytes from producing IL-2 and interferon-gamma (IFN-γ) and macrophages from producing tumor necrosis factor alpha (TNF-α) or IL-12 [22]. Resveratrol has been found to induce a dose-dependent suppression of the production of IL-1α, IL-6, and TNF-α and down-regulate both mRNA expression and protein secretion of IL-17 in vitro [23]. Dietary resveratrol supplementation is capable of improving tight junction protein zonula occludens-1, Occludin, and claudin-1 expression to reduce intestinal permeability in vivo [24,25]. Resveratrol treatment also reduced the expression of the inflammatory factors, glycation end product receptor (RAGE), NF-кB (P65) and nicotinamide adenine dinucleotide phosphate (NADPH) oxidase 4 (NOX4) and improved the renal pathological structure [26]. In addition, as the natural precursor of resveratrol, polydatin (a resveratrol glycoside isolated from Polygonum cuspidatum) significantly downregulated IL-6, IL-1β, and TNF-α expression induced by Mycoplasma gallisepticum both in vivo and in vitro, suggested that polydatin also has anti-inflammatory effect [27].

The anti-inflammatory activity of this compound was also demonstrated in a rat model of carrageenan-induced paw edema [28]. Additionally, it has been reported that resveratrol preconditioning modulates inflammatory response of hippocampus after global cerebral ischemia in rat [29]. In addition, resveratrol has been documented to be able to suppress neuro-inflammation mediated by microglia, protect neurons from inflammatory damage, and relieve inflammation of the airway caused by asthma and airway remodeling [30,31].

Heat stress may induce reactive oxygen species (ROS) production, cause antioxidant system disorders, and cause damage to the immune organs in vivo [32]. In a recent report, our research group observed that dietary resveratrol supplementation in broilers was successful in partially alleviating the detrimental effects of heat stress on the function of the intestinal barrier, by restoring the damaged villus-crypt structure, altering the mRNA expression of intestinal heat-shock proteins, secreted immunoglobulin A, and close junction-related genes and inhibiting pro-inflammation secretion [33]. Zhang et al. [34] reported that dietary supplementation with resveratrol in broilers could partly reverse the adverse effects of heat stress on the growth of immune organ by reestablishing redox status and inhibiting apoptosis. Importantly, our recent study demonstrated that resveratrol could alleviate heat stress-induced innate immunity and inflammatory response by inhibiting the activation of PRRs signaling in the spleen of broilers [35].

Lipopolysaccharides (LPS) is an essential glycolipid component of Gram-negative bacterial endotoxin which can trigger inflammatory responses to the host [36]. The addition of resveratrol has also led to decreases in inflammatory mediator expression, including prostaglandin (PG)E2, COX-2, IL-1β, IL-8, TNF-α, and monocyte chemoattractant protein-1 in BV-2 or monocyte LPS stimulation cells [37,38]. Additionally, Toll-like receptor-4 (TLR-4) expression in LPS-stimulated cells was attenuated after resveratrol pre-treatment [39]. Long-term treatment with this compound is able to increase brain defenses in aged animals against acute LPS pro-inflammatory stimuli [40]. Moreover, palmitate-induced IL-6 and TNF-α at mRNA and protein levels in C2C12 cells can also be substantially prevented by resveratrol pretreatment [41].

On the other hand, resveratrol can induce anti-inflammatory properties by suppressing the production of ROS and nitric oxide (NO). Oxidative stress caused by the accumulation of ROS plays a role in promoting inflammation in a wide spectrum of diseases, such as chronic inflammation and cancer [42]. It was found that resveratrol was able to suppress strongly the generation of NO in activated macrophages, as well as decrease strongly the amount of cytosolic inducible nitric oxide synthase (iNOS) protein and steady state mRNA levels [43]. Dietary resveratrol supplementation also can effectively eliminate free radicals, enhance the activities of SOD, CAT, and GPX [32,44]. Babu et al. [45] showed that the cytoprotective effect of resveratrol is predominantly due to mitigation of mitochondrial ROS. A recent study conducted by Kortam et al. [46] showed that resveratrol increased the liver’s antioxidant and anti-inflammatory activity against chronic unpredictable mild stress induced depression in the animal model, as explained by the normalization of total antioxidant ability, glutathione, malondialdehyde (MDA), NF-кB, TNF-α, and myeloperoxidase. Additionally, resveratrol down-regulated the expression of iNOS mRNA and protein expression in the LPS-stimulated intestinal cells in a dose-dependent manner, resulting in a decreased production of NO [39]. Similarly, resveratrol dose-dependently inhibited the expression of iNOS and IL-6 in LPS-treated RAW264.7 cells, therefore, suppressed the production of NO and the secretion of IL-6 [47].

## 4. Potential Anti-Inflammatory Pathways of Resveratrol

Many studies have reported that resveratrol regulates inflammatory response through a variety of signaling pathways, such as AA pathway [48], nuclear factor kappa B (NF-κb) [49], Mitogen-activated protein kinase (MAPK) [50], and activator protein-1 (AP-1) [51].

### 4.1. Arachidonic Acid (AA) Pathway

Inhibition of the AA pathway plays a major role, among other anti-inflammatory pathways facilitated by polyphenols [48,52]. AA is released by membrane phospholipids with the cleavage of phospholipase A2, and then metabolized by COX with generation of PGs (such as PGD2, PGE2, PGI2) and thromboxane (TX) A2 [53] (Figure 2). There is evidence that both COX forms (include COX-1 and COX-2) are significant sources of PGs [54,55]. Prostanoids produced via COX-1 exert renal homeostasis, cytoprotective, immunomodulatory, and platelet function [56], while those derived from COX-2 participates in the inflammatory response [57]. COX-2 expression is affected by many inflammatory factors, such as ultraviolet B (UVB)-radiation, TPA (12-*O*-tetradecanoylphorbol-13-acetate) or PMA (Phorbol 12-myristate 13-acetate) (a tumor promoter), and tobacco [58,59].

Resveratrol has been stated to exhibit anti-inflammatory responses and inhibit the functions of COX and hydroperoxidase, although the results have been inconsistent in various previous studies. Jang et al. [28] found that this polyphenol was able to selectively suppress the COX activity of COX-1 and this isoenzyme’s hydroperoxidase activity, thus inhibiting PG synthesis. It has also been shown that resveratrol was able to discriminate between two COX isoforms, suggesting it is a potent inhibitor of COX-1 catalytic activity, but only a poor inhibitor of COX-2 peroxidase activity, the isoform target for non-steroidal anti-inflammatory drugs [60].

In comparison, it has been demonstrated that resveratrol inhibits PGE2 synthesis by directly blocking the activity of COX-2, and suppresses the transcription of COX-2 gene without changing the amount of COX-1 in PMA-treated human mammary epithelial cells [61]. A subsequent study reported that resveratrol induced a decrease of PGs as well as COX-2 expression by reducing AA release, as well as COX-2 induction by an antioxidant action, because this phytoalexin is able to decrease O29- as well as hydrogen peroxide (H_2_O_2_) produced by PMA- or LPS-treated murine peritoneal macrophages [62]. Murias et al. [63] evaluated some methoxylated and hydroxylated resveratrol derivatives for their ability to inhibit COX isoenzymes and the results showed that hydroxylated resveratrol analogs are selective inhibitors of COX-2. Resveratrol was stated to be able to reduce the degree of colonic injury by decreasing the production of PGD2 and PGE2, as well as lowering COX-2 expression in the colon of resveratrol-treated rats compared with inflamed colon [64,65].

Resveratrol in a certain concentration can suppress human colorectal cancer and pulmonary epithelial cell (A549 cells) proliferation by inhibiting COX-2 expression [48,66]. A recent study found that this phytoalexin inhibited microphthalmia-associated transcription factor and tyrosinase activity via extracellular signal-regulated kinase (ERK) 1/2 and PI-3K/Akt pathway-mediated suppression of COX-2 [67]. Additionally, resveratrol is able to improve neuroimmune dysregulation by inhibiting of neuronal TLRs and COX-2 signaling pathways in vivo [68]. Transient transfections using deletion constructs of the COX-2 promoter as well as COX-2 promoter constructs in which unique enhancer elements are mutagenized has shown that resveratrol’s effects are mediated by a cyclic AMP response element [61].

Moreover, the literature shows that resveratrol decreases PMA-dependent PGE2 production through the down-regulation of gene transcription of COX-2 in rats indirectly by inhibiting the protein kinase C (PKC), ERK1, c-Juns, and AP-1 activities, which provides an additional mechanism for resveratrol inactivation of COX-2 [61]. A subsequent study showed that this polyphenol was able to suppress COX-2 promoter-induced transcriptional activity (include NF-κB and AP-1) and PKC activation [69,70]. Kundu et al. [71] also reported that this compound was able to inhibit oxidative stress, expression and activity of COX-2 by inhibiting NF-κB, AP-1, and Janus kinase/signal transducers and transcript activators (JAK/STAT) activation pathways in mouse skin induced by TPA. TPA/PMA is known to induce expression of COX-2 via transcriptional activation of NF-κB and AP-1 [59]. The anti-inflammatory action of resveratrol is dependent on activation of AMP-activated kinase (AMPK) and is related to inhibition of the LPS stimulation NF-κB-induced COX-2 signaling pathway in RAW 264.7 macrophages [72].

In fact, the overexpression of COX-2 isoenzymes in cancer is due to abnormal transcription and post-transcriptional control [70]. Hence, the suppression of COX-2 activity exerted by resveratrol provides a mechanistic basis for this compound’s the chemo-preventive properties [61]. Again, both COX-1 and COX-2 are potently suppressed by resveratrol [73]. Therefore, resveratrol exerts an anti-inflammatory action partly due to the suppression of COX-1, COX-2, and its antioxidant effect.

### 4.2. NF-κB Pathway

As a ubiquitous nuclear transcription factor, NF-κB modulates a wide variety of genes expression regulating inflammatory responses. The family of NF-κB/Rel transcription factor includes NF-κB1 (p50/p105), NF-κB2 (p52/p100), p65 (RelA), RelB, and c-Rel [74]. NF-κB normally exists in the cytoplasm in an inactive form, and interacts with inhibitors of κB (IκB), including IκBα and IκBβ. IKK-mediated phosphorylation of IκB is a significant step in NF-κB activation [75]. Recently, it has become very clear that there are two separate NF-κB activation pathways: the classical pathway and the alternative pathway [76,77]. The activation of NF-κB can lead to the expression of inflammatory cytokines such as IL-1, IL-6, IL-10, and TNF in LPS-stimulation cells [78]. The abilities of resveratrol inhibition of NO synthase (NOS) and the down-regulation of NF-κB activation in macrophages is higher than naringenin and naringin [43]. Furthermore, the inhibition of NF-κB activation and NF-κB-regulated gene expression by resveratrol are related to the suppression of the IκB kinase activation [79].

Resveratrol can block PMA-induced NF-κB activation [51], LPS [43], H_2_O_2_, okadaic acid, TNF-α-, IL-1β [80,81], and UVB [49]. Actually, resveratrol suppresses TNF-induced NF-κB activation in a dose- and time-dependent fashion, as demonstrated in myeloid cells, lymphoid, and epithelial cells [51]. In the same study, resveratrol was also reported to suppress phosphorylation and nuclear translocation of the p65 subunit induced by TNF-α, as well as transcription of NF-κB-dependent reporter gene. A previous study demonstrated that cells treated with LPS in addition to resveratrol exhibited suppression of NF-κB activation and phosphorylation, as well as IκBα degradation, and NF-κB subunits’ nuclear content was reduced [43]. In addition, it has been shown that resveratrol decreases the TLR-4 expression and induces the development of IL-6, NO, and iNOS induced by LPS, inhibits IκBα phosphorylation, subsequently preventing NF-κB p65 translocation from the cytoplasm to the nucleus [39,47].

Resveratrol decreases IL-1β production and also inhibits the NF-κB activation dependent by IL-1β in vitro, which regulates several signals controlling cellular survival, proliferation, and inflammatory cytokine production [22,80]. In the normal human epidermal keratinocytes, this compound has been reported to block the activation of NF-κB in a dose- and time-dependent manner caused by UVB (40 mJ/cm^2^), via inhibition of phosphorylation and degradation of IκBα as well as IKKα activation [49]. Zhang et al. [82] demonstrated that resveratrol was able to attenuate anoxia/reoxygenation injury-reduced expression of TLR4, inhibit NF-κB activation and downregulate the expression of inflammatory factors genes including TNF-α and IL-1β in vitro. Furthermore, Ren et al. [83] found that resveratrol suppressed endogenous and TNF-α-induced NF-κB activation in a dose-dependent manner. Further investigation revealed that resveratrol blocked the ubiquitination of NEMO and inhibited IκBβ-mediated NF-κB activation. Yi et al. [81] also reported that resveratrol was able to dramatically suppress the IL-1β-induced inflammation (including the COX-2, matrix metalloproteinase-1 (MMP-1), MMP-3, MMP-13, and iNOs expression) by inhibiting IL-1β-induced the degradation of IκB-α and the activation of NF-κB.

Uchida et al. [84] reported that resveratrol markedly and dose-dependently enhanced NF-κB activation triggered by pro-inflammatory cytokinesinglomerular mesangial cells. This compound was also able to suppress the ERK1/2/NF-κB pathway to inhibit oxidative stress and inflammation, and then inhibit neointimal hyperplasia after balloon injury [85]. Resveratrol can also attenuate liver fibrosis though the inhibition of the Akt/NF-κB pathways [86]. Liver fibrosis is a significant pathological change in chronic liver injury in which multiple inflammatory cytokines and signaling pathways are involved [87]. Qi et al. [88] demonstrated that resveratrol might play an important role in protection of the ethanol-induced neuro inflammatory responses by blocking TLR2-MyD88-NF-κB signal pathway. Rasheduzzaman et al. [89] pointed out that resveratrol could also be the effective TRAIL-based cancer therapy regimen by attenuating TNF-included apoptosis-inducing ligand (TRAIL) resistance, as well as inducing TRAIL-mediated apoptosis via the NF-κB p65 pathway.

In addition, resveratrol treatment improved neuroimmune dysregulation via inhibiting pro-inflammatory mediators and TLRs/NF-κB pathway [68]. In a recent study conducted by Lopes Pinheiro et al. [38], resveratrol treatment initiated substantial changes in protein acetylation and methylation patterns, indicating deacetylase induction and demethylase reduction activities that primarily affect regulatory cascades NF-кB- and JAK/STAT- mediated pathway. In summary, NF-κB is a molecular target for the treatment of inflammatory diseases. These studies suggested that NF-κB transcriptional activity suppression leads to resveratrol’s anti-inflammatory properties.

Further study of the NF-κB mechanism of resveratrol found that resveratrol can act on the corresponding cellular kinases. As a Ca_2_/CaM-dependent Ser/Thr protein kinase, Death-associated protein kinase 1 (DAPK1) has many functions to participate in several pathological and physiological processes, such as cell necrosis, apoptosis, autophagy, and innate immunity [90,91]. It has been found that DAPK can suppress NF-κB activation and expression of pro-inflammatory cytokine induced by TNF-α or LPS [92,93]. DAPK1 knockdown using siRNA abolished resveratrol-induced autophagy, while it almost did not affect the phosphorylation level of AMPK, a target of resveratrol. Resveratrol-induced autophagy in human dermal fibroblasts can be DAPK1-mediated, raising the possibility that its anti-inflammatory effects are related to its regulation of DAPK1 [94].

Resveratrol exhibits anti-inflammatory effects and immunomodulating functions via sirtuin-1 (Sirt-1) activation [95]. As a deacetylase, Sirt-1 plays a vital role in immune tolerance [96,97], and it operates by blocking the TLR-4/NF-κB/STAT pathway with decreased inflammatory factors production [6,98]. For example, it was demonstrated that Sirt1 exhibited anti-inflammatory effects by regulating NLRP3 expression in mesenchymal stem cells partially via the NF-κB signaling [99], and it suppressed Porphyromonas endodontalis LPS-induced MMP-13 expression in osteoblasts via the inhibition of NF-κB signaling [100]. Resveratrol binding to Sirt-1 enhanced its attachment to RelA/p65 substrate [101], which activated leukocytes and the pro-inflammatory cytokine pathway [102]. Resveratrol inhibited RelA acetylation via Sirt1 activation, and in turn lowered expression of genes including TNF-α, IL-1β, IL-6, MMPs, and Cox-2 induced by NF-κB [96,103]. Resveratrol has been shown to be a double inhibitor of NF-κB signaling, for the reason that treated cells are less responsive to NF-κB signaling and apoptosis initiation induced by TNF-α [51]. Moreover, resveratrol could suppress p300 expression and promote IκBα degradation, while it remains unclear whether this process occurs through the activation of Sirt1 [104]. The activation of SIRT1 by resveratrol, leading to the inhibition of phospho-p38 MAPK as well as the reduced NF-κB p65 activity [105].

### 4.3. MAPK Pathway

MAPK is activated by translocation to the nucleus, where they phosphorylate various target transcription factors, including Nrf2, NF-κb, and AP-1 [106,107]. MAPK signal transduction pathways are pivotal in many biological processes, including proliferation, differentiation, apoptosis, inflammation, and responses to environmental stresses [108]. MAPKs are a family of stress-inducible kinases, including the c-jun Nterminal kinase (JNK), extracellular-signal regulated kinase (ERK), Big MAP kinase (BMK), and p38 (Figure 3) [109,110]. Among these, the p38 MAPK is activated by several pro-inflammatory stimuli such as oxidative stress, UBV, and inflammatory cytokines [111,112].

It has been reported that resveratrol suppresses COX-2 expression in mouse skin ex vivo by suppressing the activation of ERKs and p38 MAPK pathways induced by PMA [64]. The inhibitory effects of this compound probably involve suppression of the p38 MAPK-cytosolic phospholipase A2-AA-TxA2-[Ca^2+^]_i_ cascade and NO/cyclic GMP activation, then lead to suppression of phospholipase C and/or PKC activation [113]. Resveratrol in combination with pemetrexed shows a synergistic cytotoxic effect, accompanied with the reduction of protein levels of phospho-p38 MAPK and ERCC1, as well as a DNA repair capacity [114]. Resveratrol has also been reported to suppress invasion and migration of pancreatic cancer cells by blocking the ERKs and p38 MAPK induced by hyperglycemia-driven ROS [115]. In vivo studies reported that maternal dietary resveratrol addition decreased the levels of T cell receptor genes, MAPK signal transduction pathways in weaning piglets, and it alleviated weaning-associated diarrhea and intestinal inflammatory disorders in porcine offspring [116]. Moreover, many studies have shown that this compound could potentially induce apoptosis and in vitro ROS accumulation through p38 MAPK signaling pathways [117,118,119]. Yang et al. [120] also reported that resveratrol was able to reduce ROS accumulation, inflammation, and angiogenesis both in vivo and in vitro, then displaying preventive effects for rheumatoid arthritis (RA). Additionally, it can alleviate the nervous inflammatory responses after injury via decreasing the extracellular levels of gliotransmitter, as well as blocking p38 MAPK activation [121,122]. Moreover, the inhibition of MAPK signaling pathways probably contributes to the anti-inflammatory effects of resveratrol.

Resveratrol is able to suppress the inflammatory response though blocking the phosphorylation protein expression of p65 and IκB from the NF-κB signaling as well as phosphorylation of p38 and ERK from MAPK signaling under mastitis conditions [123]. Our recently study also demonstrated that dietary resveratrol supplementation in broilers could inhibit inflammatory response by inhibiting NF-κB, MAPK, and PI3K/AKT signaling under heat stress condition [35]. Wang et al. [124] pointed out that this phytoalexins could block the signaling cascades of NF-κB p65 and MAPKs in vivo by downregulating the mRNA expression of genes involved in NF-κB and MAPKs in LPS-induced liver and lung in rats, and inhibiting the dynamic changes of proteins and phosphorylated proteins which include IκBα, NF-κB p65, JNK, ERK1/2, ERK5, and p38 MAPK from tissue cytoplasm to nucleus.

### 4.4. AP-1 Pathway

AP-1 is another transcription factor typically comprised of one member each from the Jun (c-Jun, JunB and JunD) and Fos (c-Fos, FosB, Fra1 and Fra2) families [125]. AP-1 regulates an array of cell processes, including proliferation, differentiation, inflammation, and apoptosis [126]. AP-1 can be activated by various extracellular stimuli [125], and resveratrol can block PMA- or TNF- induced activation of AP-1-mediated gene expression (Figure 3) [51,61]. Resveratrol can suppress IL-8 production in U937 cells induced by PMA at both mRNA and protein levels [127], suggesting that the block on IL-8 gene transcription by this compound is due partly to suppression of AP-1 activation. The downregulation of AP-1 activity may contribute to resveratrol’s anti-proliferative activity in A431 cells [128]. Although studies have shown that resveratrol can easily inhibit COX-2 activity directly, the indirect inhibitory effect of reducing COX-2 expression appears to be more significant after inhibiting AP-1 [129].

In addition, suppression of NF-κB correlated with inhibition of AP-1, and the anti-carcinogenic and anti-inflammatory effects of resveratrol can therefore be partly due to blocking both NF-κB and AP-1 activation as well as related kinases [51]. Resveratrol provides chemical protection against cancer through the NF-κB and AP-1 pathways [129]. This has been shown in mouse skin models in which abnormal NF-κB and AP-1 activities lead to skin tumors [130]. Furthermore, in vitro study conducted by Donnelly et al. [131] found that the inhibitory effects of this phytoalexin are greater than those of the glucocorticosteroid dexamethasone, due to the suppression of transcription of NF-κB- and AP-1-, as well as the protein-dependent cyclic adenosine monophosphate response element binding.

### 4.5. Antioxidant Defense Pathways

Resveratrol exhibits strong antioxidant activity, mainly via the control of major antioxidant enzymes and block on DNA damage by free radicals. The anti-inflammatory activity of this compound may be due to the antioxidant properties by inhibiting pro-inflammatory signaling pathways (Figure 3). Many studies have demonstrated that resveratrol can reduce H_2_O_2_-dependent oxidative damage in calf thymus DNA [117,132], as well as in several cancer cell lines [133,134]. As a known marker of oxidative DNA damage, the levels of 8-oxo-7,8-dihydroxy-29-deoxyguanosine decrease with the addition of resveratrol in vivo [135]. Moreover, resveratrol is able to alleviate intestinal injury and dysfunctions though improving oxidative status and inhibiting inflammatory responses in rats under heat stress [136].

High levels of NO are produced by iNOS in inflammation, and the inhibition of iNOS expression, might play a major role in anti-inflammatory responses [137]. It has been shown that resveratrol can protect IPEC-J2 cells from oxidative damage by stimulating the Nrf2 pathway [138]. Additionally, resveratrol upregulates the protein expression of Nrf2 and HO-1 signaling pathway, thus attenuating oxidative stress and inflammatory response by hypoxic-ichemically induced in neonatal rats [139]. In addition, resveratrol also inhibits the activation of MAPKs and reduces the production of inflammatory mediators by activating SIRT1/AMPK and Nrf2 antioxidant defense pathways [140]. Therefore, one of the anti-inflammatory effects of resveratrol may be explained by the suppression of the antioxidant properties.

## 5. Summary and Perspectives

Multiple lines of compelling evidence indicate that resveratrol has a promising role in the prevention and treatment of many autoimmune and inflammatory chronic diseases, such as inflammatory, neurological, and multiple cancers. This phytoalexin was demonstrated to modulate many cellular and molecular mediators of inflammation, but the molecular mechanisms of polyphenol are complex and involve multiple signal transduction pathways, and have not been fully elucidated.

Therefore, future research should focus on: (1) further evaluating the compound during clinical trials and improving oral absorption efficiency; and (2) elucidating the underlying mechanisms of resveratrol action in several physiological conditions, in order to make this compound a cutting-edge therapeutic strategy for the prevention and treatment of a wide variety of chronic diseases.

## Figures and Tables

**Figure 1 molecules-26-00229-f001:**
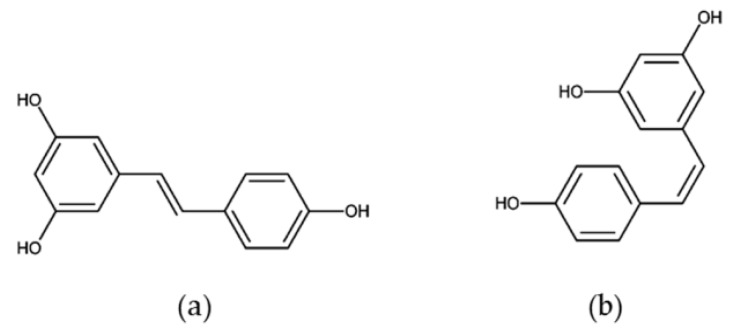
Structure of trans-resveratrol (**a**) and cis-resveratrol (**b**).

**Figure 2 molecules-26-00229-f002:**
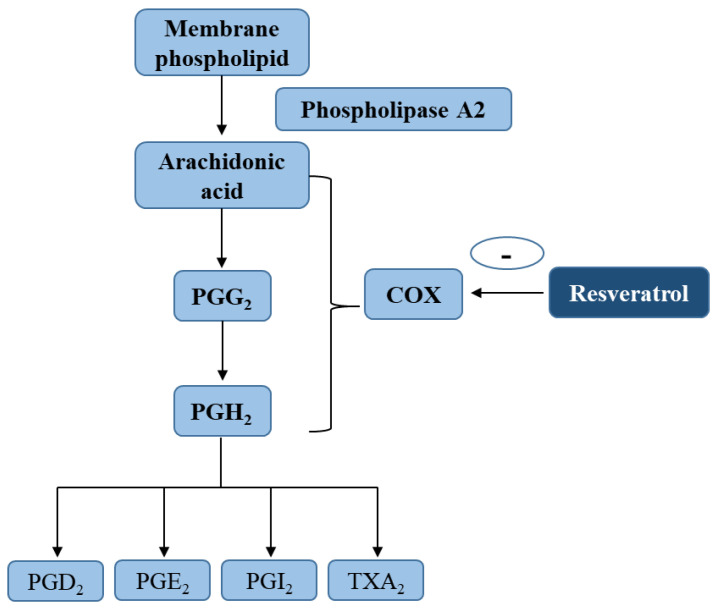
Inhibition of arachidonic acid metabolic pathway by resveratrol. Key abbreviations: COX, cyclooxygenase; PGD2, prostaglandin D2; PGE2, prostaglandin E2; PGI2, prostaglandin I2; TXA2, thromboxane A2.

**Figure 3 molecules-26-00229-f003:**
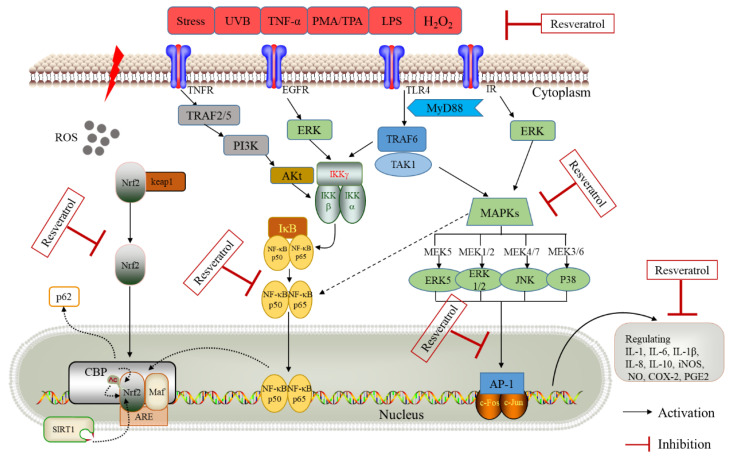
Schematic presentation of possible signaling cascades of resveratrol in suppression inflammatory response. Key abbreviations: UVB, ultraviolet B; TNF-α, tumor necrosis factor alpha; TPA, 12-*O*-tetradecanoylphorbol-13-acetate; PMA, Phorbol 12-myristate 13-acetate; H_2_O_2_, hydrogen peroxide; LPS, lipopolysaccharide; TLR4, Toll-like receptor 4; MyD88, myeloid differentiation factor 88; TRAF6, tumor necrosis factor receptor associated factor 6; MAPKs, mitogen-activated protein kinases; IκB, inhibitor kappa B; IKK, inhibitor-κB kinase; NF-κBp65, nuclear factor kappa B 3; IL, interleukin; NO, nitric oxide; iNOS, inducible nitric oxide synthase; COX-2, cyclooxygenase-2; PGE2, prostaglandin E2.

**Table 1 molecules-26-00229-t001:** The content of resveratrol in some food products.

Plants	Content
Mulberries	5 mg of resveratrol per 100 g
Lingonberries	3 mg per 100 g
Cranberries	1.92 mg of resveratrol per 100 g
Red currants	1.57 mg of resveratrol per 100 g
Bilberries	0.67 mg of resveratrol per 100 g
Blueberries	0.383 mg per 100 g
Peanuts	1.12 mg of resveratrol per 100 g
Pistachios	0.11 mg of resveratrol per 100 g
Fresh grapes	0.24 to 1.25 mg per cup (160 g)
Red grape juice	0.5 mg per liter

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
