# Peer review of "Anti-Inflammatory Action and Mechanisms of Resveratrol"

_molecules, 2021, doi:10.3390/molecules26010229_

Round 1

Reviewer 1 Report

The paper offers an up-to-date overview on the multiple anti-inflammatory actions of resveratrol. It merely focuses on cellular mechanisms which can be involved in several diseases.

The manuscript is well-structured.

I have only some minor points:

  • Title of fig.1: structure refers to resveratrol. Either the authors add the stilbene chemical structure (for instance as panel a, and then res as b and c), or they modify the title in Structure of trans-resveratrol (a) and cis-resveratrol (b).
  • Line 83, pag 2: polydatin has been demonstrated a very interesting molecule in preventing inflammation. I would suggest to state here that it also the natural precursor of resveratrol.
  • Paragraph 4: I would suggest to move the first sentence (lines 135-137) of subheading 4.1 before it and after paragraph 4 as introductory sentence.
  • Line 152: “resveratrol was able to distinguish” I would suggest to substitute distinguish with another term.. maybe is able to act differently on? What do the author mean with distinguish?
  • Line 167: please add epithelial to A549
  • Figure 2: You should address the title to resveratrol. For ex.: Inhibition of arachidonic acid metabolic pathway by resveratrol.
  • Paragraph 4.2: Line 201-202. I would mention all the 5 members of the NFkB family and eliminate and so on.
  • Paragraph 5: As it represents the final discussion, I would replace this phytoalexin (line 376) with resveratrol, and resveratrol (378) with this phytoalexin.

Reviewer 2 Report

Reviewed manuscript entitled „Anti-Inflammatory Action and Mechanisms of Resveratrol” submitted by Tiantian Meng et al. is very interesting and summarized information about anti-inflammatory properties of resveratrol.

In my opinion the Authors ought to consider the addition of the table with the content of resveratrol in some food products. Also information about the amount of resveratrol needed for anti-inflammatory activity in human organism could be interesting for readers.
